# A systematic review of resilient performance in defence and security settings

**Marc Vincent Jones[1]\*, Nathan Smith[2], Danielle Burns[1], Elizabeth Braithwaite[1], Martin Turner [1], Andy McCann[1], Lucy Walker[1], Paul Emmerson[3], Leonie Webster[4], Martin Jones[4]**

**1** Department of Psychology, Manchester Metropolitan University, Manchester, England, **2** Centre for Trust, Peace and Social Relations, Coventry University, Coventry, England, **3** Cervus AI, Salisbury, United Kingdom, **4** Human and Social Sciences Group, Defence Science and Technology Laboratory, Salisbury, United Kingdom

\* marc.jones@mmu.ac.uk

**Data Availability Statement:** All relevant data are within the manuscript and its Supporting Information files.

**Funding:** This work was funded by the Human Social Science Research Capability (HSSRC)

## Abstract

A narrative systematic literature review was conducted to explore resilient performance in defence and security settings. A search strategy was employed across a total of five databases, searching published articles from 2001 onwards that assessed performance and optimal function in relation to resilience, in defence and security personnel. Following narrative synthesis, studies were assessed for quality. Thirty-two articles met inclusion criteria across a range of performance domains, including, but not limited to, course selection, marksmanship, land navigation, and simulated captivity. Some of the key findings included measures of mental toughness, confidence, and a stress-is-enhancing mindset being positively associated with performance outcomes. There was mixed evidence for the predictive value of biomarkers, although there was some support for cortisol, dehydroepiandrosterone sulfate (DHEA-S) and neuropeptide-y (NPY), and vagal reactivity. Interventions to improve resilient performance were focused on mindfulness or general psychological skills, with effects generally clearer on cognitive tasks rather than direct performance outcomes in the field. In sum, no single measure, nor intervention was consistently associated with performance over a range of domains. To inform future work, findings from the present review have been used to develop a framework of resilient performance, with the aim to promote theoretically informed work.

## Introduction

Defence and security personnel operate in Volatile, Uncertain, Complex, and Ambiguous (VUCA) settings. To perform well in VUCA settings, a range of physical, behavioural and psychological competencies are required. Examples include problem solving, being able to learn new roles, tasks and technologies, and demonstrating interpersonal, cultural and physical adaptability [1]. To help personnel maintain and optimise performance in these and other similar competency areas, it is important to understand the antecedents that explain variability in

research fund, project HS 1.025 Psychological Resilience to Maximise Human Performance. The award was made to MVJ, NS, MT, EB. The funders (MJ, LW, PE) did play a role in reviewing and editing in the preparation of the manuscript. They provided comments on structure and writing style.

**Competing interests:** The authors have declared that no competing interests exist.

their execution. A factor that has been implicated in such performance in the military, but as yet remains poorly understood, is psychological resilience [2,3].

In defence and security domains, resilience has typically been studied in the context of traumatic stress by examining its buffering role in experiences of mental (ill) health [4–7]. Findings from this body of research have highlighted trajectories of resilient responding [8] and the role of personality traits, values, coping strategies and social resources in mental health experiences of military personnel [9–12]. Nonetheless, a recent meta-analysis found psychological resilience was not strongly predictive of mental health and functioning in military personnel, with the heterogeneity of the measures of resilience, and health and functioning outcomes hindering clear conclusions [13].

However, psychological resilience is not solely about mental health. It also relates to performance, particularly in psychologically demanding VUCA settings. When performance matters, the demands of the situation can generate a stress response which can potentially impair function, such as co-ordinated motor control, decision-making, or attention. Situational demands that could cause stress include uncertainty, danger (which can be either physical or psychological such as a threat to esteem), and the requirement for effort [14,15], which are often present in the role of defence and security personnel. Psychological resilience is reflected in positive adaptation to adversity or stress [16]. This adaptation can help maintain sufficient performance levels to execute important competencies [1], and achieve a successful outcome. It is the association between psychological resilience and performance that is the focus of the present review. For the purposes of the present work, resilient performance is defined as: *The maintained or improved execution of competence under situational duress.* Thus, resilient performance is a behaviour, not a personal trait or biopsychosocial process. Rather, personal traits and biopsychosocial processes, plus external factors (e.g. social support) are thought to predict and underpin resilient performance in a situationally specific manner. With this in mind, resilient performance is self-referent in that it is defined by relative (individual) change in performance from base levels. An individual who is able to execute their competencies successfully under demanding conditions, with equal to or exceeding, usual performance, is thought to demonstrate resilient performance.

Outside of the military, and beyond the focus on mental health, psychological resilience has been the target of numerous performance-related studies [17–19]. This work indicates that the role of resilience may move past simply protecting against degradations in mental health, to also protecting against degradation in performance and increasing function. As an illustration, Fletcher and Sarkar [20] proposed a grounded theory of resilience that linked stressor exposure to optimal sporting performance via a number of key factors including individual differences in personality, motivation, and resources such as confidence and perceived social support. Together, these factors were linked to challenge stress appraisals and facilitating performance under stress. In relation to high performance more broadly, the importance of experience and learning, perceptions of control and flexibility, and adaptability were highlighted as other factors associated with resilience [21,22]. Further recent advances in this area have provided a more structured organisation of factors likely to contribute to resilience when performing in high stress settings, like those faced by defence and security personnel. Work by Brown and colleagues [23], highlighted a number of personal and environmental enablers (e.g., resilient qualities, psychological skill use and social support) and processes (e.g., challenge appraisal and psychological need for satisfaction) that contribute to both performance and wellbeing in demanding situations. While further work [24] has hinted at a potentially instructive link to biological processes that could underpin resilient performance, (e.g., Cortisol, Dehydroepiandrosterone [DHEA]). Collectively, existing work on resilient performance in high stress contexts suggests that there are likely to be both enabling factors and processes that determine maintenance and optimal performance under stress. Importantly, these enablers

and processes seem to be captured by a constellation of biological, psychological and social dynamics. Indeed, exploring the biological, psychological, and social aspects of resilience draws on a longstanding tradition of studying stress and performance from a biopsychosocial basis [14,25,26].

Consistent with this biopsychosocial view, Kalisch, Cramer [27] recently proposed the concept of a dynamic resilience network, in which psychological resilience is a dynamic context specific construct, dependent on person-environment interactions. Conceptualised with a focus on mental health, resilience can be understood by the dynamics in the interconnections between physical, affective, cognitive, and social nodes. Stressor exposure can lead to deterioration in the different nodes, which if left unchecked can spread through the network. For example, deterioration in the physical node (e.g., sleep) can impact upon affective responses (e.g., increased negative mood), which might disrupt cognitive capability (e.g., lack of concentration) and damage social relations (e.g., being short-tempered with peers). Resilience would be demonstrated by quick recovery in the affected nodes and limited disruption through the network (e.g., maintaining a positive mood state). According to Kalisch, Cramer [27], so-called resilience factors (e.g., personality traits), act as moderators and protection to the network to foster resilient responding under conditions of stress. The model provides a coherent link between important resilient outcomes (i.e., the nodes), resilient factors (i.e., potential antecedents) and a consideration of these dynamics over time. This temporal view can be applied to resilient performance in VUCA settings. Chronic exposure to stress may need to be managed over extended periods to avoid adverse performance-related impacts. Whereas acute stress exposure may require personnel to transition quickly from periods of low activation to engagement in psychologically and physically demanding tasks. Resilience is likely to play an important role in all of these situations; which defence and security personnel encounter multiple times across their career lifespan.

Prior research studies conducted with defence, security, and law enforcement personnel document several measurable performance related metrics linked to resilient function, such as passing selection courses, marksmanship, and navigation [22,28,29]. Researchers have also examined the role of various individual difference factors including personality and motivation [30,31], psychological skills such as imagery and activation regulation [32], and biomarkers like cortisol, neuropeptide-Y (NPY), dehydroepiandrosterone (DHEA) and vagal tone [33], in an attempt to explain who is likely to respond resiliently to stressor exposure and able to maintain performance. Currently, these disparate yet related studies have not been synthesised in a way that might inform the biopsychosocial factors underpinning resilient performance and thus contribute to methods for monitoring and optimising the function of defence and security personnel. With this in mind, the present work builds on recent contributions by van der Meulen, van der Velden [13] and examines resilience with a specific focus on its relationship with performance (as opposed to mental health) in defence and security settings.

The overarching aim of the present work was to conduct a narrative systematic review of resilient performance in defence and security settings. The review was organised around core research questions which included:

1. What theories and models have been used to study factors associated with resilient performance in defence and security settings and other high performance domains?

2. What measures and metrics have been used to study factors associated with resilient performance in defence and security settings and other high performance domains?

3. How have programmes designed to bolster resilient performance in defence and security settings and other high performance domains been developed and what is the evidence for their effectiveness?

4. What is the quality of papers published in the field?

The intended outcome of the review was to inform recommendations on the methods that could be used to measure resilient performance, to evaluate the effectiveness of related training programmes designed for defence and security personnel and to develop a framework of resilient performance for future research.

## Methods

As part of the present review, research studies on the mechanisms, measures, and approaches used to evaluate and to train resilient performance in defence and security settings were identified and assessed. A focus was placed on understanding performance under conditions of stress (psychological and physical), with resilient performance demonstrated by an individual who is able to execute their competencies successfully under demanding conditions, such that situational performance is equal to, or exceeds, their usual performance. In contrast, an individual who is unable to execute their competencies successfully under demanding conditions, such that situational performance is below their usual standard, and not sufficient to complete the task demands successfully is thought to not be demonstrating resilient performance. Resilience was defined as the maintained or improved execution of competence under situational duress, and a resilience training environment was defined as one which aims to enhance or maintain resilient performance in a high-stress environment with a focus on biopsychosocial markers and targets. A high-stress environment was defined as those that are VUCA, where there is potential for harm and effort is required.

### Literature search strategy and selection criteria

A systematic literature search for articles published in English from January 2001 to October 2020 was conducted, to identify primary research studies that have investigated the relationship between psychological resilience and performance in high-stress environments. Subsequently an updated search was performed, searching for articles published from October 2020 to January 2022, to maintain currency for the completed review. Target populations included military personnel, security/defence personnel, and those who embark upon expeditions or space travel. A pilot search was initially conducted including a broad target range of populations (e.g., elite sports people, first responders). Based on this search the heterogeneity of papers was such that we delineated the search terms to the target populations of military personnel, security/defence personnel, expeditions/space travel to ensure a more focused and relevant population for our literature review. The decision to include articles published from 2001 onwards was based on the timing of when psychological resilience, beyond the broader study of stress, emerged as a specific focus in scientific research, and a contemporary view of the military environment, which has been substantially shaped by operations that have taken place since 2001 [34,35].

**Search strategy.** In accordance with the PRISMA guidelines [36] the search strategy was developed for this systematic review and the search was conducted (Commenced 6th of August, 2020). Electronic literature searches were carried out using the title field in the following online databases: MEDLINE via PubMed, PsychINFO, Scopus, and Web of Science. Grey literature was searched for in the Defence Technology Information Centre (DTIC). Reference lists of included studies and relevant review articles identified through the search were also checked for relevant articles, including military reports. The search terms used were: (Resilien* OR stress* OR cope* OR "mental toughness" OR challenge OR threat OR pressure OR risk OR fortitude OR thriv* OR flourish* OR grit OR hardiness OR robust*) AND (Perform* OR

behavio* OR response* OR biopsychosocial OR psych* OR intervention OR program OR training OR develop* OR accuracy) AND (Military OR army OR airforce OR navy OR "defence personnel" OR "defence staff" OR "defence employees" OR "SOF" OR "special forces" OR "security personnel" OR "security staff" OR "security employees" OR space OR expedition OR police).

Inclusion and Exclusion criteria: Articles were identified that met the following criteria: original, peer-reviewed studies including the following designs: theoretical/conceptual papers, cohort studies, cross-sectional studies, case-control studies, qualitative studies, and prospective and longitudinal designs. Articles were required to be published in English, from 2001 onwards, and relate to performance and optimal function in relation to resilience. Articles were excluded if they focused on participants aged 18 and under, or undergoing basic and entry level training programmes in defence and security environments. Pre-search exclusion criteria also included wrong population, wrong paper type such as review articles, editorials, conference abstracts, book chapters, not published in English, and papers that did not measure performance.

**Data extraction.** *Paper selection process.* Titles and articles retrieved by the online search strategy were reviewed by a member of the research team. If titles appeared to meet the inclusion criteria citations were sent to an online reference manager to be stored for abstract review; if it was not clear from the title whether the article subject matter was of interest a pre-screen of the abstract was performed. Duplicates of papers retrieved were then removed and abstracts of the remaining papers were reviewed against the inclusion and exclusion criteria by the two members of the research team. Full texts were obtained for the final selected articles. See Fig 1 for a flow diagram of the search process.

*Data extraction.* Following the search strategy, papers identified for further review were read by at least two members of the research team, and every paper was considered for quality and relevance by the research team as a whole during meetings. Data were extracted from eligible articles and entered into a pre-defined data-extraction sheet. The following data were extracted from each article: paper details (authors, year), study design (sample size, participant details such as age and area of occupation, recruitment method and procedure), details of the measure of performance, theoretical approach, markers of resilience, details of the training program (if relevant), key findings. Inconsistencies were resolved in consensus meetings attended by the whole project team.

**Literature search outcome.** The original literature search yielded 3047 articles. 1546 duplicates were deleted, and 1415 were discarded as irrelevant/ineligible based on titles and abstracts. The remaining 131 articles were assessed for inclusion by a member of the research team using the full text, and a further 99 were excluded. This resulted in 32 articles that met inclusion criteria for the systematic review. The updated search retrieved a total of 415 titles, after which 274 remained, following duplicate removal. Following this 254 were discarded as irrelevant or ineligible leaving 20 titles being assessed for full inclusion, however none met the full criteria for inclusion in the review. Due to the heterogeneity of papers and small numbers of articles retrieved to individual categories the use of meta-analysis techniques was deemed not appropriate for the current review, therefore a solely narrative approach was applied to the current systematic review.

**Assessing study quality and relevance.** For each included article, two members of the research team evaluated the study quality using the Mixed Methods Appraisal Tool (MMAT), a tool designed for the appraisal stage of systematic mixed studies reviews [37]. The MMAT uses two initial screening questions, and a further five items to determine quality, and yields a score of 0 to 5, where higher scores indicate higher quality research. Members of the research

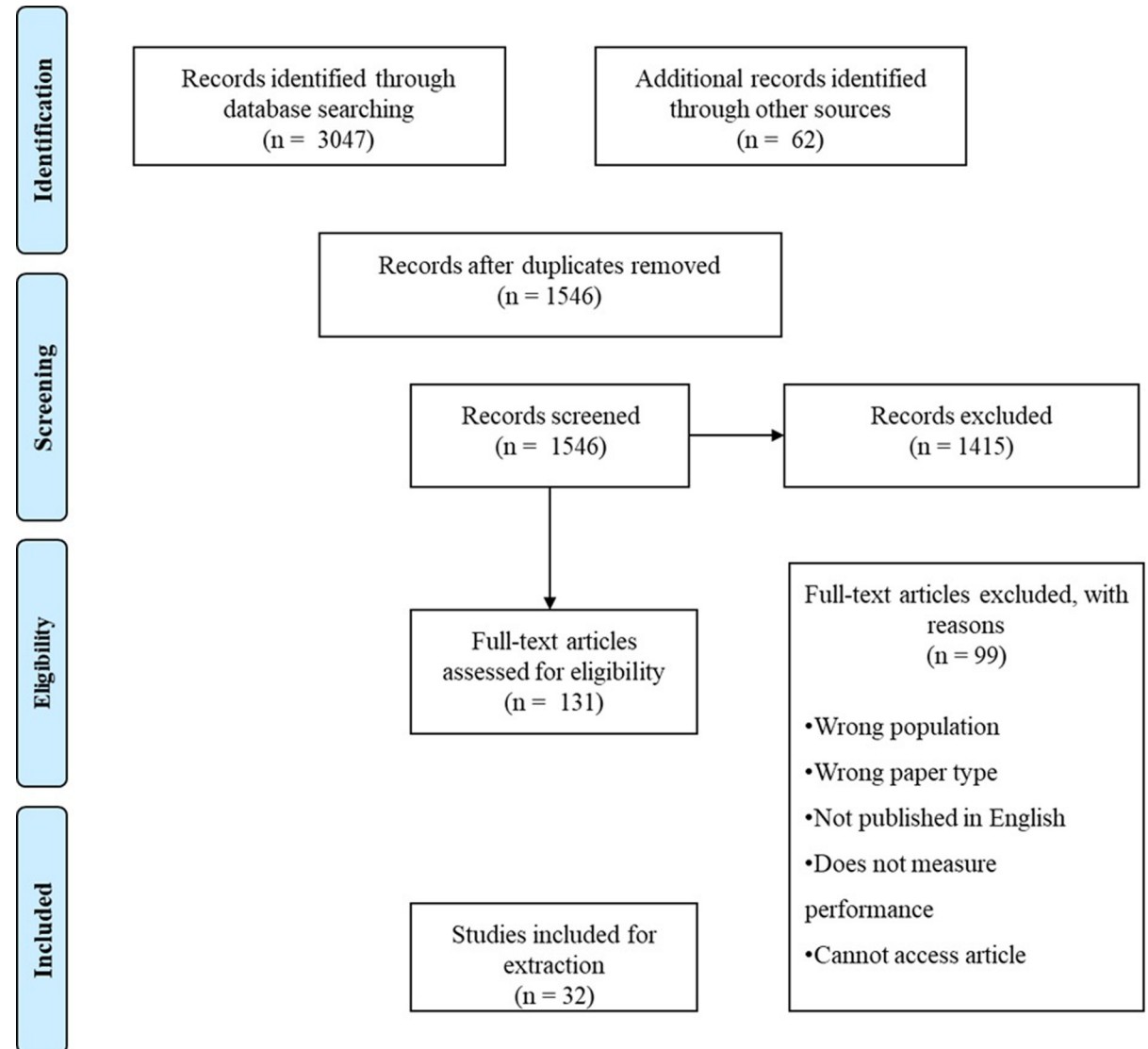

**Fig 1. PRISMA flow chart.** Flow chart explaining the search process for the systematic review.

team met to reach consensus on ratings and the papers' applicability to the review based on the inclusion and exclusion criteria.

## Results

### Overview and quality of included studies

The search retrieved 32 that satisfied inclusion criteria for data extraction and narrative synthesis, with one paper retrieved comprising of three individual studies, making a total of 34 studies included in the review. Studies were conducted internationally, with a range of defence and security personnel. A summary of the included papers, including country or origin and sample details and extraction of study characteristics is presented in Table 1. Out of the included studies [34] the majority were awarded a quality rating of three (56%) while 32% were awarded a higher rating of four. Lower proportions of paper were awarded the lower

**Table 1. Summary of studies included in the systematic review.**

| Author and year | Sample characteristics | Study Design | Measure of Performance | Theoretical approach | Markers of resilience | Key findings | MMAT Quality Score |
|---|---|---|---|---|---|---|---|
| Bartone et al 2008 [38] | $n$ = 1138 US Special Forces Selection and Assessment candidates $M$ = 25.41 years old Male Army | **Observational** Participants completed self-report measures of hardiness upon entry to SF selection, which was analysed as a potential predictor of course completion. The sample consists of four class cohorts for whom complete hardiness and graduate data were available. | **Successful SGC** Graduate vs non-graduate—US Army SF assessment and selection course. | No specific theory of resilience applied. | **Psychological** Psychological hardiness—DRS **Physiological** N/A. | Psychological hardiness was a significant predictor of success in the course (graduation). For each 1-point increase in hardiness scores, the probability of graduation increased by approximately 3.3%. | 4 |
| Beal et al 2010 [39] | $n$ = 824 US Special Forces Selection and Assessment candidates $M$ = 26 years old Sex not reported Army | **Observational** SFSA candidates completed a battery of cognitive ability tests, physical fitness measures, and perseverance tests (including a measure of grit) prior to completing SFAS performance events. There measures were used in statistical analyses to predict performance outcomes. | **Successful SCG** Pass/Fail | No specific theory of resilience applied. | **Psychological** Perseverance—Duckworth Grit Scale **Physiological** N/A. | Self-reported perseverance on the grit scale was significantly associated with selection, whereby a higher level of grit was associated with a better chance of selection. Of the four grit subscales, three were significantly associated with selection (perseverance of effort, brief grit and ambition), whereas the consistency of interest subscale was not associated with selection. | 4 |
| Campbell et al. et al. 2017 [40] | $n$ = 30 US Marine Corps Sergeants Age not reported Male Army | **Experimental (non-randomized)** The aim of this study was to design and validate a virtual reality training tool designed to improve small unit leader decision making in the field. Participants completed a pre-test Situational Judgement Test (SJT). The sample was then split into the control and experimental groups and received approximately 8-hours of training. The control group (made up of eight leaders) received three scenarios created by the | **Cognitive** Decision-making was measured using pre and post-test competency assessed by SJT ratings, compared to subject matter experts. Decision-making expertise level was assessed during the field study by instructors using the Behaviourally Anchored Rating Scale (BARS) to evaluate trainees in Key Performance Areas (KPAs). | Maintenance of equilibrium under adversity (Bonnano 2004, Masten & Narayan 2012 [41,42]) Lazarus's theory of stress and coping (Lazarus 1966) [43]. Thriving under adversity (Epel 1998) [44]. | **Psychological** Anxiety—State Trait Anxiety Inventory (STAI) Resilience—Connor-Davidson Resilience Scale (CD-RISC) **Physiological** N/A | Results suggest, but do not definitively prove, that the STAR-DM simulation training packages: 1. Can induce a significant physiological stress response, even in experienced Marines; 2. Can improve decision-making performance during training; and 3. Can improve decision-making performance in stressful field exercises. | 3 |

*(Continued)*

**Table 1.** (Continued)

| Author and year | Sample characteristics | Study Design | Measure of Performance | Theoretical approach | Markers of resilience | Key findings | MMAT Quality Score |
|---|---|---|---|---|---|---|---|
| | | simulation lab at School of Infantry–East (SOI-?E) and the experimental group (five leaders) received five scenarios of the STAR-?DM SLTPs. In the STAR-?DM group, each participant interacts with the VBS2 scenario to make decisions and enact responses to events. The instructor provides overall feedback on mission and decision event performance. All participants then completed a post-test SJT and a series of field exercises. | | | | | |
| Canada et al 2018 [45] | n = 64 US ARMY Special Operation Forces M = 31.1 years old Male Army | **Experimental (non-randomised)** This study compared the shooting performance between special forces operators who had completed the THOR3 program vs those who had not. De-identified archival data used to examine participant performance in the Special Forces Urban Combat stress shoot. | **Shooting** Advanced Urban Combat stress shoot—a dynamic shooting task with a range of psychological and physiological demands. Time, in seconds, was collected and comparisons made between users and nonusers in four performance categories: raw time, total time, Positive Identification (PID) time, and penalty time. | No specific theory of resilience applied. | N/A | There were no statistically significant differences in shooting performance between the two groups. However, it appears that Operators exposed to the THOR3 program may have a greater opportunity for success when engaging targets under physical and psychological duress. Having a higher level of physical performance and knowledge of mental skills training methods may provide advantages to Operators performing in a stress shoot or similar tasks. | 3 |
| Eid and Morgan 2006 [46] | n = 56 Norwegian Navy M = 24.8 years old Mixed (91% male, 7% female) Navy | **Observational** Participants self-reported dissociative states at time point 1 (after exposure to a mild stress) and time point 2 (within 2-hours of completing survival training). Participants also self-reported psychological hardiness after time point 1. | **Applied** Independent Expert Raters (IER) assessed performance in an interrogation task (low stress component, time point 1) and in a mock POW experience (high stress component, time point 2). Participant performance was rated between 0 (poor) | No specific theory of resilience applied | **Psychological** Dissociative experiences—CADSS Hardiness—Norwegian translation of short-form DRS **Physiological** N/A | Symptoms of peritraumatic dissociation were negatively and significantly related to performance. Cadets who reported more symptoms of dissociation after the relatively mild stress at time point 1 reported more symptoms of dissociation at time point 2. Cadets who exhibited greater symptoms of dissociation did not perform as well as their peers who were not dissociators. | 3 |

*(Continued)*

**Table 1.** (Continued)

| Author and year | Sample characteristics | Study Design | Measure of Performance | Theoretical approach | Markers of resilience | Key findings | MMAT Quality Score |
|---|---|---|---|---|---|---|---|
| | | | and 5 (very well) on two dimensions: 1) Verbal performance —adherence to strict disclosure protocols; and 2) Nonverbal performance—control of body language and posture An independent rating was produced for average verbal, nonverbal and total performance. | | | | |
| Farina et al 2019 [47] | *n* = 800 US Special Forces Selection and Assessment candidates At least 20 years old Male Army | **Observational** Demographic & psychological predictors (grit & resiliency) collected at beginning of SFAS with fasted blood sample, remainder of physical & psychological predictors collected during SFAS. | **SCG** Successful selection. **Applied** Army Physical Fitness Test (APFT). Two timed runs and two loaded road marches Obstacle course: sum of points of each obstacle completed. Land navigation task: number of coordinates successfully located. | No specific theory of resilience applied. | **Psychological** Grit—Duckworth short Grit scale Resilience—CD-RISC **Physiological** Cortisol, DHEA-S, testosterone, SHBG, and CRP.) | Higher cortisol was associated with higher probability of selection & correlated with higher resiliency/grit scores. Participants who self-reported a higher level of grit and resilience were more likely to be selected. Basal serum physiological markers weakly predicted selection and were weakly associated with behavioural assessments. Lower CRP and higher cortisol and SHBGH predicted selection. Higher CRP was associated with lower fitness test scores and slower road march time. SHBG correlated with better performance on pull-ups, land navigation, obstacle course, and the fitness test. Testosterone was correlated with faster run and road march times. DHEA-S correlated with lower resilience scores, and DHEA-S, epinephrine, and norepinephrine correlated with worse performance on several physical events. | 3 |
| Fitzwater et al 2018 [32] | *n* = 186 Para recruits & Parachute Regiment corporals, UK **Recruits:** *M* = 21.13 years old **Corporals:** *M* = 28.44 years old Male Army | **Experimental (non-randomised)** This study used a quasi-experimental design to examine the impact of mental skills training on hardiness and performance in British Army Para recruits. | **Applied** P-Company score (maximum of 70 points) based on their performance in seven events, where a maximum of 10 points are awarded. P-company staff determine score for each event. 2-mile loaded run & negotiation of steeplechase course. | No specific theory of resilience applied. | **Psychological** Mental toughness—Military Training Mental Toughness Inventory (MTMTI) Psychological skills—Test of Performance Strategies (TOPS-2) Within-group leadership climate -Differentiated Transformation Leadership Inventory (DTLI). **Physiological** N/A. | From time point 1 to 2, there was a significant increase in in observer-rated mental toughness in the experimental group, whereas there was no change in contro+H11l. Individual performance was significantly higher during P-Company for experimental group when controlling for fitness & leadership climate. Experimental group had higher | 5 |

*(Continued)*

**Table 1.** (*Continued*)

| Author and year | Sample characteristics | Study Design | Measure of Performance | Theoretical approach | Markers of resilience | Key findings | MMAT Quality Score |
|---|---|---|---|---|---|---|---|
| | | 10 platoons were included in the study, and whole platoons were assigned to either the intervention (n = 5) or control (n = 5) conditions on a rotating basis. Data gathered at 2 time points, 3- weeks apart. | | | | overall pass rates during P-company, although non-significant | |
| Gayton et al 2015 [48] | n = 95 Australian Army SF Unit applicants M = 26.9 years old Male Army | **Observational** Participants completed self-report measures on their first day of assessment, which was assessed against their subsequent selection (pass/fail). | **Successful SGC** Australian Army (SF applicants were categorised as follows: 1. Those who were withdrawn on initial day of assessment and did not start 3-week selection (Did Not Start; DNS) 2. Those who started but were withdrawn from selection course (Did Not Finish, DNF) 3. Those who completed selection but were withdrawn following a negative recommendation (Not Recommended, NR); and 4. Those who completed selection and passed onto further training (Pass). | Psychological hardiness as a framework for understanding of resilient functioning that includes cognitive, emotional, and behavioral qualities (Bartone et al. et al. 1998, 2008 [38,49]) | **Psychological** Psychological Hardiness—DRS-15 (short form) **Physiological** N/A | Examination of hardiness scores indicated successful applicants were similar to unsuccessful applicants on measures of self-reported hardiness. Hardiness scores were not significantly associated with top-ranked strength of persistence. | 3 |
| Gepner et al 2019 [28] | n = 20 Elite combat unit, Israel Defence Forces (IDF) M = 20.1 years old Male Army | **Observational** Soldiers completed intense field training simulating a sustained military operation. Blood samples were taken 3 hours postprandial, then participants completed a cognitive function assessment, military-specific physical test (200m gauntlet run) and then attended the shooting range to assess static & dynamic marksmanship. Soldiers were from the same unit, garrisoned on base, performed same activities & ate the same meals. | **Shooting** Static shooting—participant lay prone Dynamic shooting—Deliver fire to a fixed target moving from position to position in upright position. Each shot that hit the target was considered accurate. Time to complete shooting and accuracy were both recorded. **Cognitive** Modified version of the original Serial Sevens Test. | No specific theory of resilience applied. | **Psychological** N/A **Physiological** Serum concentrations of interleukin-10 (IL10) and tumour necrosis factor-a (TNF-a). Plasma concentrations of BDNF and GFAP. | Significant inverse correlations were noted between TNF-a concentrations and dynamic shooting accuracy. Trend noted in association of TNF-a concentrations and both static shooting accuracy and target engagement speed. Trends also noted between IL-10 concentrations and dynamic shooting performance, no significant correlation was noted with static shooting. BDNF concentrations were significantly correlated with the Serial Sevens performance and number of correct answers. Trend towards inverse association between static shooting performance and GFAP concentrations. | 3 |

(*Continued*)

**Table 1.** (*Continued*)

| Author and year | Sample characteristics | Study Design | Measure of Performance | Theoretical approach | Markers of resilience | Key findings | MMAT Quality Score |
|---|---|---|---|---|---|---|---|
| Gucciardi et al 2015 [40] | *n* = 115 Australian Defence Force selection course *M* = 27.16 years old Male Army | **Observational** Participants completed a survey on the first evening of the 6-week selection course. The selection test consisted of a 6-week selection course specifically designed to test suitability for SF recruitment. Candidate performance was continuously monitored by instructional staff. In total, 50 candidates (43%) passed the course. | **Successful SGC** Australian Defence Force applying for Special Forces Fail or pass | Mental toughness (s the defining attribute that enables one to thrive in demanding situations (Jones & Moorhouse, 2007; Weinberg, 2010 [50,51]). | **Psychological** Mental toughness—Mental Toughness Inventory (developed in study 2) Psychological hardiness—15-item Norwegian DRS Self efficacy—8-item new general self-efficacy scale. **Physiological** N/A | Mental toughness significantly predicted selection test outcome, even when hardiness and self-efficacy were considered. That is, those participants who self-reported as more mentally tough were more likely to pass selection. The specific facets of commitment, control, challenge and self-efficacy did not predict performance outcome. | 4 |
| Gucciardi et al 2021 [22] | *n* = 122 Australian Special forces selection course *M* = 27.56 years old All male except 1 female Other | **Observational** Participants provided hair sample and completed Mental Toughness Scale prior to completing the 3-week selection course. | **Successful SGC** Applying for unit Fail or pass. | Mental toughness as the defining attribute that enables one to thrive in demanding situations (Jones & Moorhouse, 2007; Weinberg, 2010 [50,51]). | **Psychological** Mental toughness—Mental Toughness Inventory **Physiological** Hair cortisol | There was a small-to-moderate association between mental toughness and perseverance in the selection course, accounting for chronological age and accumulated stress (assessed via hair cortisol levels). | 4 |
| Hardy et al 2010 [52] | *n* = 484 UK Royal Marine Commandos *M* = 20.1 years old Male Army | **Observational** Participants completed battery of tests prior to Royal Marine Commando training. Performance was measured at the end of training, as a pass or fail of the course. | **Successful SGC** Civilian recruits training to become a Royal Marine Commando. Completion or withdrawal. | No specific theory of resilience applied. | **Psychological** Resilience—Six independent items developed in line with conceptualisation Self-confidence—modified Trait Sport Confidence Inventory. **Physiological** N/A | Participants who self-reported a high level of resilience and self-confidence were more likely to complete the Royal Marine Commando training course. | 3 |
| Jensen et al 2020 [29] | *n* = 203 US Marine Corps applicants *M* = 22.7 years old Male Marines | **Experimental (randomised)** Participants were US Marine Corps applicants who took part in a randomised trial and were randomised to one of three groups during training: Mindfulness-Based Fitness Training (MBFT), | **Applied** Phase 1: the assessment of a hike, Reconnaissance Physical Assessment Test (RPAT), Physical Fitness Test (PFT), land navigation, and the phase 1 test. Phase 2: the amphibious skills test and a final average score. | No specific theory of resilience applied. | **Psychological** N/A **Physiological** Insulin-like growth factor (IGF-1) Cortisol Adrenaline | Overall, the results suggest that incorporating mental skills training into military training may lead to improvements in cognitive performance. There were mixed results between groups on operational performance tasks with the MST groups (i.e., MMFT & GMST) tending to perform better than TAU the more time participants had with MST instruction. | 4 |

(*Continued*)

**Table 1.** (Continued)

| Author and year | Sample characteristics | Study Design | Measure of Performance | Theoretical approach | Markers of resilience | Key findings | MMAT Quality Score |
|---|---|---|---|---|---|---|---|
| | | General Mental Skills Training (GMST) or Training as Usual (TAU). Marine corps training was split into three phases: intervention groups received additional MBFT or GMST training during each phase. Physiological assessments were taken at 4 time points during phase 3 (baseline, pre-ambush, ambush, and post-ambush). Operational performance was collected across all three phases. Cognitive skills assessment completed during phase 2 and phase 3. | Phase 3: hike, patrol 1, 2, and 3, and a communications test. Each skills test/assessment are graded on a sliding scale, with the maximum score being 100 points. **Cognitive** SART test Date/time recall test Coordinates recall test Plot accuracy/time test Facial recognition accuracy/time test "Kim's game". | | | During ambush, the differences among groups were especially pronounced for measures of information processing that one would expect MST to enhance: coordinates recall, plot time, and plot accuracy, with improvements ranging from 24.7 to 87.9% for the MST conditions when compared to TAU. | |
| Jha et al 2015 [53] | $n$ = 124 (**M8T/M8D** = 40; **NTC** = 24; **CIV** = 60) US Army & US Marine Corps Reserves **M8T** = 26.7 years old; **M8D** = 25.8 years old; **NTC** = 27 years old; **CIV** = 20.44 years old **M8T/M8D** = Male; **NTC** = Not reported; **CIV** = All male except 3 female Army | **Experimental (randomised)** US active-duty soldier cohorts were randomly assigned by unit to either the 8-hour MMFT variant that emphasized didactic content (M8D) or the 8-hour MMFT variant that emphasized MT practices (M8T). All participants were tested before (T1) and after (T2) an 8-week training period. | **Cognitive** SART 1) Attentional performance errors (A', errors of commission); 2) individual RT-variability (i.e., the intra-individual coefficient of variation (ICV); and 3) subjective reports of-mind wandering elicited by probes. | No specific theory of resilience applied. | **Psychological** N/A **Physiological** N/A | At T2, both MT groups had higher A' scores and self-reported being significantly (M8D) or marginally (M8T) more 'aware' of their attention compared to NTC. In addition, a direct comparison of the two MT groups revealed that M8T had higher A' and lower ICV relative to M8D, suggesting fewer attentional lapses and greater attentional stability in the M8T group. As predicted, CIV A' scores remained stable over time while the NTC group's scores significantly declined from T1 to T2. Like NTC, M8D also significantly degraded over time, suggesting that participating in this MT course did not sufficiently protect against performance costs over time. In contrast, M8T's A' scores remained stable from T1 to T2. These results suggest that attentional performance, which degrades over the pre-deployment interval, is better protected by M8T vs. M8D. | 2 |

*(Continued)*

**Table 1.** (Continued)

| Author and year | Sample characteristics | Study Design | Measure of Performance | Theoretical approach | Markers of resilience | Key findings | MMAT Quality Score |
|---|---|---|---|---|---|---|---|
| | | | | | | These results suggest that MT focused on in-class training exercises more so than on in-class didactic instruction may promote cognitive resilience by protecting attentional capacities put at risk by high-demand intervals. | |
| Jha et al 2020 [54] | $n$ = 180 US Army military base PE: $M$ = 23.57 years old, ME: $M$ = 23.31 years old, NTC: $M$ = 23.48 years old Male Army | **Experimental (mix of randomisation)** Three companies were assigned to receive training from Master Resilience Trainer-Performance Experts naive to Mindfulness, but trained to deliver for the study (PE) or a ME trainer. The PE group had two subgroups, taught by different PE trainers. A fourth company was assigned to NTC group, non-randomized. Soldiers were assessed in the week before the training period (week 0, T1), following the training period (week 5, T2) and 4-weeks following T2 (week 10, T3). | **Cognitive** SART Working Memory Delayed-Recognition Task with Affective Distracters | No specific theory of resilience applied. | **Psychological** N/A **Physiological** N/A | While task performance declined over the high-demand military field training interval for all participants, the PE group showed less decline when compared to the NTC group, as well as the ME group. PE group was estimated to decline less compared to the NTC group by 0.08 units of A′ in the SART from T1 to T2, which represent roughly eight fewer missed SART target trials. Only directional but not significant differences between the PE and NTC groups in the amount of attentional change from T1 to T3, suggesting that the attentional benefits were not completely maintained across the entire study interval (10-weeks in total). PE vs ME comparison on SART was comparable. PE group was estimated to decline less compared to the NTC group by 3.07% accuracy in the WMDA task from T1 to T3, which represents roughly two fewer incorrect WMDA trials on average. these results suggest that not only was PE-delivered MBAT protective against decline relative to the NTC group, but it was also protective relative to ME-delivered MBAT. | 4 |
| Johnsen et al 2013 [55] | $n$ = 178 Norwegian Military Personnel $M$ = 19.9 years old All male except 3 females Army | **Observational** Soldiers participated in a 250 km long ski march as the final part of the selection course for entry into the border patrol rangers forces tasked to protect the border between Norway and Russia. | **Successful SGC** Success or failure on ski march. | No specific theory of resilience applied. | **Psychological** Hardiness—DRS-15-R Coping and self-appraisal—Visual Analog Scales (VAS). Sensation seeking—Arnett Inventory of Sensation Seeking (AISS). **Physiological** N/A | Total hardiness score was the only significant predictor of success on the ski march when controlling for physical fitness, nutrition and sensation-seeking. | 5 |

*(Continued)*

**Table 1.** (Continued)

| Author and year | Sample characteristics | Study Design | Measure of Performance | Theoretical approach | Markers of resilience | Key findings | MMAT Quality Score |
|---|---|---|---|---|---|---|---|
| | | Participants completed measures before the ski march & visual analogue scales each evening once camp had been built across the 9-days | | | | | |
| Landman et al 2016 [56] | $n = 59$ Dutch police officers including regular and specialist Arrest Unit (AU) **P1 Pre-AU:** $M$ = 29.1 years old; **P1 AU:** $M$ = 33.6 years old; **P2 Pre-Au:** $M$ = 29.4 years old; **P2 AU:** $M$ = 30.6 years old Male Police | **Experimental** Participants were regular and specialist AU, and officers who wanted to join the AU (pre-AU officers). In phase 1 of the study officers completed a survey, and in Phase 2 officers completed two shooting tasks (low and high pressure tasks, counterbalanced). | **Shooting** Shooting performance consisted of shot accuracy and the number of incorrect shooting decisions. Shot accuracy was operationalised as the number of target hits on armed trials and expressed as a percentage of the total number of shots on armed trials. Incorrect shooting decisions were defined as the number of unarmed trials in which the officers fired (i.e., false positives) and was expressed as a percentage of the total number of unarmed trials. Movement time: measured with eye-tracking glasses. Gaze behaviour: fixations on the opponent's gun. | No specific theory of resilience applied | **Psychological** Anxiety—STAI, anxiety thermometer Behaviour—Behavioural Inhibition System and Behavioural Activation System scales (BIS/BAS). Self-control—Action Control Scale Impulsivity—Dickman Impulsivity Inventory Sensation seeking— Sensation Seeking Scale Rating Scale of Mental Effort. **Physiological** Mean heart rate. | Regression analyses showed that state anxiety and shooting performance under high pressure were first predicted by AU experience and second by certain personality traits. Results suggest that although personality traits attenuate the impact of high pressure, it is relevant experience that secures effective performance under pressure. Personality traits that seemed to be adaptive included low sensitivity to threat, high self-control strength and affinity for thrill and adventure Results suggest that maintaining performance in high-pressure situations is a skill that is sensitive to practise. | 3 |
| Lieberman et al 2002 [57] | $n = 68$ Navy SEAL trainees $M$ = 23.9 years old Male Navy | **Double-blind placebo-controlled** Prior to hell week, participants completed measures of demographics and training on the cognitive tests (baseline measures). Caffeine doses of 100, 200, 300 milligrams (mg) or placebo were administered randomly to volunteers. Hell week began on Sunday night, | **Cognitive** Scanning visual vigilance Four-choice visual reaction time Matching-to-sample test Repeated acquisition test **Shooting** Prone firing position with sandbags to support the rifle. Following a "ready signal" and an interval of 1–10-seconds (randomly varied), a red light emitting diode | No specific theory of resilience applied. | **Psychological** Mood states—Profile of Mood States (POMS) **Physiological** Salivary caffeine levels | Almost all cognitive and mood measures were substantially degraded during Hell week compared with baseline measures. Caffeine produced significant beneficial, dose-related effects for visual vigilance, visual reaction time, and motor learning. There were no effects of caffeine on short term spatial working memory, pattern recognition, or marksmanship. Caffeine consumption improved self-reported mood and sleepiness. | 4 |

*(Continued)*

**Table 1.** (Continued)

| Author and year | Sample characteristics | Study Design | Measure of Performance | Theoretical approach | Markers of resilience | Key findings | MMAT Quality Score |
|---|---|---|---|---|---|---|---|
| | | and the following Wednesday night at 2130 caffeine/placebo was consumed. Volunteers were almost totally sleep-deprived for 72-hours prior to administration of the test substances. At 2230 tests were administered in a classroom, followed by a meal and return to physically demanding training. 8- hours after administration participant returned to the classroom to repeat the tests and the post-test questionnaire. | was illuminated indicating the subject could start shooting. Volunteers then fired at the target as quickly and accurately as possible. Assessed distance from centre of mass, shot group tightness, number of missed targets, and sighting time | | | | |
| Lieberman et al 2009 [58] | *n* = 15 US Army Ranger School Not reported Male Army | **Within-subjects repeated measures** Participants cognitive function and rectal temperature (for core body temp) was assessed on three separate occasions during three standardised cold exposure tests conducted in an environmental chamber: 1) Immediately (within 2-hours) after they had completed the final field exercise of Ranger training (Day 1), 2) After a short (48hour) rest and recovery period; and 3) After a long (108-day) recovery period (Day 109) after they had returned to their regular duties. The stressors of ranger school include fatigue, chronic sleep loss, nutritional deprivation and psychological stress. Only eight volunteers returned for the final assessment. | **Cognitive** Computer: Visual vigilance , four-choice reaction time test. Pen & paper: Pattern recognition , symbol-digit substitution , word-list learning , grammatical reasoning | No specific theory of resilience applied. | **Psychological** Mood states—POMS **Physiological** Core body temperature | Cognitive performance improved over time following the completion of Ranger school. All mood states improved over time following completion of Ranger school. Effects of acute cold stress were observed for the visual vigilance cognitive test— correct hits and reaction time were impaired by cold stress. There were also effects of acute cold stress in impairing the pattern recognition test, and on increasing the tension mood state. | 3 |

*(Continued)*

**Table 1.** (Continued)

| Author and year | Sample characteristics | Study Design | Measure of Performance | Theoretical approach | Markers of resilience | Key findings | MMAT Quality Score |
|---|---|---|---|---|---|---|---|
| Lieberman et al. et al. 2016 [59] | *n* = 60 US Army Special Forces on SERE course *M* = 26.92 years old Male Army | **Observational** Baseline measures (non-stressed) were collected with participants after the last training session: saliva three per day. Practice 1, blood draw, cognitive/mood tests & heart rate. Two testing sessions during captivity phase, each immediately after an interrogation exercise: saliva, blood draw, cognitive/mood tests & heart rate. High concordance at baseline testing but more variation during captivity phase, as testing could not obstruct training objectives. | **Cognitive** Psychomotor Vigilance Test Match-to-Sample N-back Grammatical Reasoning Test (adapted from Baddeley) | No specific theory of resilience applied. | **Psychological** Mood states—Profile of Mood States (POMS) **Physiological** Cortisol Testosterone NPY DHEA-S Adrenaline Noradrenaline Prolactin | SERE school was associated with: substantially degraded mental and psychological functioning as demonstrated by significant and robust effects on multiple cognitive tasks and mood scales, increased levels of cortisol and suppressed release of anabolic hormone testosterone, substantially elevated adrenaline, noradrenaline and heart-rate in absence of strenuous physical activity. Response times and other measures of performance on all cognitive tests administered, Grammatical Reasoning, N-back, Match-to-Sample and Psychomotor Vigilance, were significantly degraded from baseline levels. | 3 |
| Meland et al 2015 [60] | *n* = 40 (training = 25; control = 15) Norwegian helicopter pilots **Training:** *M* = 35 years old; **control:** *M* = 40 years old Sex not reported Airforce | **Experimental (non-randomised)** This study sought to determine if MT has a measurable impact on stress and attentional control as measured by objective physiological and psychological means. The effects of 4-month MT on salivary cortisol and performance on two computer-based cognitive tasks were tested on a military helicopter unit exposed to a prolonged period of high workload. MT participants were compared to a wait list control group on levels of saliva cortisol and performance on a go–no go test and a test of stimulus-driven attentional capture. | **Cognitive** Go/No-Go task (Sustained Attention to Respond Task; SART) Stimulus driven attentional capture task (Attentional Capture Task, ACT). | No specific theory of resilience applied | **Psychological** Mindfulness -Five Facet Mindfulness Questionnaire (FFMQ; Dundas et al, 2013 [61]) **Physiological** Cortisol | MT participants compared to the control group had a larger pre to post increase in high- and low-cortisol slopes, and decrease in perceived mental demand imposed by the go–no go test. MT program increased the observation and description aspects of mindfulness, indicating that the restorative effects of MT came through increased exposure to present-moment experience and a more relaxed and flexible mind, less vulnerable to habitual responding. MT resulted in slower RT, which authors argued as indicative of greater attentional focus and purposeful responding. MT alleviates some of the physiological stress response and the subjective mental demands of challenging tasks in a military helicopter unit during a period of high workload. | 3 |

(*Continued*)

**Table 1.** (Continued)

| Author and year | Sample characteristics | Study Design | Measure of Performance | Theoretical approach | Markers of resilience | Key findings | MMAT Quality Score |
|---|---|---|---|---|---|---|---|
| | | Saliva sample for cortisol assay were collected at three time points: waking and 30-?minutes post-waking, and bed-time.<br>Pre and post-intervention data collection. | | | | | |
| Morgan et al 2001 [62] | $n = 44$<br>US Army soldiers on SERE course<br>$M = 27.8$ years old<br>Male<br>Army | **Observational**<br>Participants were divided into two subgroups: those who were sampled at baseline and recovery (n = 23) and those who were sampled at baseline and immediately after exposure to interrogation stress (n = 21). Subjects were randomly assigned to the groups. Baseline blood draws were taken 5-days before the stress test, and saliva samples were collected. Participants also completed the Cloninger TPQ. 21 blood samples were collected immediately after interrogation, and 23 recovery samples collected 24-hours later. | **Applied**<br>The stress test consisted of a captivity experience with physical and mental stress, and sleep and food deprivation. Performance scored by survival school instructors.<br>Full performance data not available.<br>Score is a reflection of how interesting their behaviour was to the interrogator, to assess mental clarity. | No specific theory of resilience applied | **Psychological**<br>Distress—Subjective Units of Distress Scale (SUDS)<br>Dissociation—CADSS<br>Personality traits—Cloninger TPQ<br>**Physiological**<br>Plasma catecholamines<br>NPY<br>Cortisol | There were significant and positive correlations between interrogation performance, and free cortisol in response to interrogation stress (r = .45, p<0.006) and NPY after interrogation (r = 0.58, p<0.006). This indicates that those participants who had a larger cortisol and NPY response to interrogation stress, performed better during the interrogation. There was a negative and significant association between dissociation and interrogation performance (r = -.49, p<0.04), indicating that those with lower dissociation performed better. | 3 |
| Morgan et al 2004 [63] | $n = 25$<br>American Navy personnel on SERE course<br>$M = 25$ years old<br>Male<br>Navy | **Observational**<br>5-days before stress exposure (POWC phase of SERE course), baseline saliva samples were taken at 4pm on the second day of classroom activities, followed by blood plasma collection. Baseline saliva samples were taken again at 7.45am, and participants self-reported dissociation. Participants then completed the experiential phase of survival training in a mock prisoner of war camp, | **Applied**<br>Survival school instructors performed an objective appraisal of observable military-relevant performance of each participant during the POWC phase of survival school. These performance assessment scores are part of the survival school program and are not available to the public. The overall rating score, however, is designed to reflect how well a participant in training is able to demonstrate specific behaviours and | No specific theory of resilience applied | **Psychological**<br>Dissociation—CADSS<br>**Physiological**<br>Salivary cortisol<br>DHEA-S | The DHEA-S-cortisol ratio was significantly and positively associated with performance during the interrogation task. There was also a negative correlation between the stress-induced levels of salivary cortisol and interrogation performance, indicating that increased salivary cortisol was associated with poorer performance. Additionally, there was a negative association between stress-induced dissociation scores and interrogation performance (r = -0.51, p<0.01), indicating that those with fewer dissociative symptoms performed better. | 4 |

(*Continued*)

**Table 1.** (Continued)

| Author and year | Sample characteristics | Study Design | Measure of Performance | Theoretical approach | Markers of resilience | Key findings | MMAT Quality Score |
|---|---|---|---|---|---|---|---|
| | | which included food and sleep deprivation. Blood and saliva samples were collected following the interrogation stress between 1630 and 1700. | problem solving abilities while experiencing acute stress. The performance ratings are scored on a scale that ranges from 0 (no skills demonstrated) to a maximum score of 4 (excellent demonstration of skills). | | | | |
| Morgan III et al 2007a [64] | $n = 20$ U.S. Navy personnel on SERE course $M = 25$ years old Male Navy | **Observational** High frequency (HF) spectral power and heart rate data were collected during the didactic phase of the survival school training and 1 week prior to the stressful confinement phase of training. | **Applied** Mock captivity event (details classified). Class instructors (professional military interrogators) created a "captivity performance score." The score was the sum of observed, classified, target skills that survival students are expected to demonstrate. No participant engages in physical activity other than standing or leaning against the wall during the interview test. All participants are food deprived for 12-hours prior to the interview test. | No specific theory of resilience applied | **Psychological** N/A **Physiological** Vagal tone | Decreased vagal tone (as measured by HF spectral power) predicted superior performance during the captivity interview. Heart rate was not associated with performance, but had a significant inverse correlation with HF power. | 3 |
| Morgan III et al 2007b [33] | $n = 18$ Special Forces Underwater Warfare Operations Course $M = 27$ years old Male Navy | **Observational** Baseline vagal tone data were collected the day prior to the start of the didactic portion of the training program in the same manner as in Morgan III et al, 2007a | **Applied** Participants were completing the Combat Diver Qualification course (CDQC). For the final assessment, students were placed in the water approximately 3miles off shore at night. They were required to navigate to a target point and not resurface until it was reached. **Cognitive** Written test performance was based on an 85-item multiple choice test. | No specific theory of resilience applied | **Psychological** N/A **Physiological** Vagal tone | Decreased baseline vagal tone predicted superior performance in a highly stressful underwater navigation exam at the end of the course. Heart rate failed to show any significant effect on the performance variables or the predictive power of HF spectral power. There was no effect of vagal tone on the written exam. | 4 |
| Morgan III et al 2007c [64] | $n = 16$ US Navy personnel on SERE course $M = 23.88$ years old Male Navy | **Observational** Participants were assessed at 16:30 on the second day of the classroom phase, 1-week prior to the captivity training phase, same procedure as Morgan III et al, 2007a. | **Applied** As detailed in Morgan III et al. et al. 2007a. | No specific theory of resilience applied | **Psychological** N/A **Physiological** Vagal tone | Reduced vagal tone was predictive of superior performance in the mock captivity task. | 3 |

*(Continued)*

**Table 1.** (Continued)

| Author and year | Sample characteristics | Study Design | Measure of Performance | Theoretical approach | Markers of resilience | Key findings | MMAT Quality Score |
|---|---|---|---|---|---|---|---|
| Różański et al 2020 [65] | $n$ = 15 Polish Special Forces $M$ = 33.1 years old Male Army | **Observational** Participants were Polish special forces completing a 48h survival training course combined with sleep deprivation. Blood samples, divided attention and handgrip strength were measured at 3 time-points: before training (1), 24-? hours after training (2) and 48-?hours after training (3). | **Cognitive** Divided attention—90 second computer task. | No specific theory of resilience applied | **Psychological** N/A **Physiological** Creatine Kinase (CK) activity The Lipid peroxidation index (LOOS). Superoxide dismutase (SOD) and glutathione peroxidase (GPx). | Survival training combined with sleep deprivation did not cause oxidative stress or muscle tissue damage. Moreover, the soldiers did not show any deterioration in psychomotor abilities. On the contrary, there was a slight improvement in the divided attention index. | 4 |
| Smith et al 2020 [66] | $n$ = 146 US Navy SEAL training Not reported Male Navy | **Observational** Participants were completing Navy SEAL phase 1 training, in which candidates undergo 7-weeks of intense physical and mental training. Participants completed all measures prior to the commencement of the training, follow up surveys were completed prior to the 4th week of training and upon completion of the training or removal from the class. The present study focuses on the predictive power of measures collected at baseline. | **Applied** Completion times for an obstacle course, a 4 mile run, and a two-mile swim **Successful SGC** Completion of first phase; graduate or non-graduate Persistence measured as time spent in training. | No specific theory of resilience applied | **Psychological** Stress Mindset Measure—Failure is Enhancing Mindset —Willpower Mindset—This included whether a candidate was committed to BUD/S, the candidate's optimism for completing training, and whether he had a mentor who prepared him for training. Social Desirability Measures **Physiological** N/A | Stress mindset predicts outcomes over and above a number of other baseline characteristics, including demographics (highest education level and mother's education), fitness (Body Mass Index) and self-report individual differences (social desirability and optimism for success). Failure-is-enhancing mindsets predicted worse outcomes in this setting. Candidates with a failure-is enhancing mindset have slower obstacle course times, drop sooner from training, and have higher rates of dropout than their peers. There was no evidence that willpower mindsets predicted performance, persistence, or success in training, improved performance on obstacle course times, last longer in the program, and are rated more positively by peers and instructors. These candidates do not show significantly greater completion rates. | 5 |
| Szivak et al 2018 [67] | $n$ = 20 US Navy & Marine Corps on SERE training. 18–35 years old (no mean reported) Men Navy | **Observational** Participants participating in the SERE course were tested at three time points during the course: baseline (first day of SERE training, T1), a stress assessment (10d after baseline, T2) and a recovery assessment (24h after stress, T3). All testing was conducted between 1800 and 2200. | **Applied** Handgrip strength Vertical jump test | No specific theory of resilience applied | **Psychological** N/A **Physiological** Serum cortisol Serum testosterone Plasma neuropeptide Y Adrenaline Noradrenaline Dopamine | Exposure to SERE stress resulted in significant increases in plasma adrenaline noradrenaline, dopamine and cortisol concentrations, with a concomitant reduction in serum testosterone. No significant elevations in plasma NPY were observed at T2; however, a significant reduction in NPY was observed at T3. | 3 |

*(Continued)*

**Table 1.** (Continued)

| Author and year | Sample characteristics | Study Design | Measure of Performance | Theoretical approach | Markers of resilience | Key findings | MMAT Quality Score |
|---|---|---|---|---|---|---|---|
| Taylor et al 2007 [68] | *n* = 19 US Navy personnel on SERE course *M* = 21.5 years old Male Navy | **Observational** Baseline salivary samples were obtained over the course of 2 consecutive days, 1–3 weeks prior to the start of survival training, and were taken again during the stressful captivity phase of the SERE course. Participants also self-reported symptoms of dissociation and distress. | **Applied** Scored during the Stressful Captivity phase of the SERE course—students were graded on several observed survival target skills during both high- and low- intensity captivity-related challenges. | No specific theory of resilience applied | **Psychological** Dissociation—CADSS Distress—Impact of Events Scale-Revised (IES-R) **Physiological** (DHEAS Cortisol | DHEAS and the DHEAS-cortisol ratio were inversely related to overall performance during the high-intensity challenge, but both were positively associated with performance during the low-intensity challenge. Cortisol reactivity to stressful captivity tended to exacerbate, and DHEAS tended to ameliorate, the subsequent impact of stressful, captivity-related events. No relationship between dissociation and performance. | 3 |
| Taylor et al 2014 [69] | *n* = 335 US Special Forces undergoing SERE training *M* = 25.0 years old Male Army | **Observational** Most participants (57.2%) were general soldiers, whereas 41.3% were Special Forces. Participants took part in a mock-captivity event as part of their SERE training course. | **Applied** Mock-captivity event performance assessed by trained observers. Score is the sum of observed, classified, target skills that survival students are expected to demonstrate during a high-intensity mock-captivity challenge. | No specific theory of resilience applied | **Psychological** Dissociation—CADSS **Physiological** N/A | Dissociation was related to poorer objective military performance in all survival trainees. This pattern remained whether dissociative states were characterized as spontaneous, deliberate, facilitative, or debilitative. Spontaneous and deliberate dissociators, however, did not differ on military performance, nor did those individuals who appraised dissociative states as facilitative versus debilitative to stress coping, | 3 |
| Tingestad et al 2019 [70] | *n* = 219 Canadian Armed Forces *M* = 35.5 years old Males = 133, Females = 86 Army | **Observational** Blood samples were taken from participants prior to completion of the exercise tests. | **Applied** Participants completed the following tasks: Sandbag fortification, Escape to cover, Picking and digging, Pickets and wire carry, Stretcher carry and vehicle extrication. Total performance was calculated by ranking each individual score and giving it a percentile score (best performance = 100, lowest performance = 1) based on rank order Participants also completed: Aerobic test (20m shuttle run), upper body strength (grip strength test), and abdominal strength (maximal prone plank). | No specific theory of resilience applied | **Psychological** N/A **Physiological** Cortisol Adiponectin CRP INF-γ TNF-α IL 1β IL-2 IL-6 IL-8 IL-18. | The results showed that higher CRP values were associated with lower total performance scores, a slower picking and digging time, lower aerobic capacity and shorter plank time. A positive association between IL-2 values and grip strength was also observed. Adiponectin values were positively associated with plank time, but negatively associated with grip strength. | 4 |

(*Continued*)

**Table 1.** (Continued)

| Author and year | Sample characteristics | Study Design | Measure of Performance | Theoretical approach | Markers of resilience | Key findings | MMAT Quality Score |
|---|---|---|---|---|---|---|---|
| Wolf et al 2016 [71] | n = 21 Battlefield Airmen development course M = 25 years old Male Airforce | **Observational** Measured serotonin in blood samples prior to and during training for a total of six blood draws. | **Successful SGC** Completed basic training and taking part in Battlefield Airmen development courses (Graduates/Failures/Self-initiated eliminations (SIE)) **Cognitive** The continuous performance task tests reaction time processing, and decision making. Results were measured as continuous memory reaction time, mathematics processing mean reaction time, and rapid decision making mean reaction time. | No specific theory of resilience applied | **Psychological** Mood states—POMS **Physiological** Serotonin | Subjects with increased serotonin levels were more likely to SIE. Participants with higher levels of serotonin, confusion-bewilderment, depression-dejection, and vigour-activity were more likely to quit, while those with higher levels of friendliness and tension-anxiety were more likely to graduate or fail. No significant statistical relationship between serotonin and mean RT, memory, decision making or mathematical performance. | 3 |
| Yao et al 2016 [72] | n = 66 (**stress** = 38, **control** = 28) Chinese special police cadets M = 23.98 years old Male Police | **Experimental (randomised)** Participants formed a Control group & stress group. Participants were tested in single sessions and not allowed to watch other participants performing the tasks. Participants in the stress group were required to walk on an aerial rope ladder bridge. Although the chosen stressor is a standard part of cadet training, neither the stress group nor the control group had ever received such training before participating in the present study. | **Cognitive** Participant completed Go/No-Go task two minutes after receiving the intervention. | | **Psychological** Single item measures of: • Subjective nervousness • Subjective fear • Subjective control **Physiological** Heart rate | Greater heart rate increases during the rope bridge task were positively correlated with post-error slowing and had a trend of negative correlation with post-error miss rate increase in the subsequent Go/No-go task. Results suggested that stronger autonomic stress responses are related to better post-error adjustment under acute stress in this highly selected population and demonstrate that, under certain conditions, individuals with high-stress jobs might show cognitive benefits from a stronger physiological stress response. | 3 |

ratings of two (3%) and the highest rating of five (9%). Ratings indicate that the majority of papers had moderate quality, suggesting improvements could be made in future research to address this.

**Psychological theories and models used to study resilience in defence and security settings.** The collation of theories and models from the included papers is challenging for various reasons. The majority of studies retrieved did not articulate explicit underpinning, or

overarching, resilience theory in the development of study rationale. Therefore, the included studies offer little by way of theoretical consensus. For the small selection of papers that did include a theoretical underpinning [22,40,48,73], theories included the maintenance of equilibrium under adversity [41,74], stress and coping [75], thriving under adversity [44] and psychological hardiness [38]. The result is surprising given the availability of resilience theory in the general and performance-specific literature, such as the compensatory model (resilience as a factor that neutralises exposures to risk), the challenge model (a risk factor can enhance adaptation), and the protective factor model (interaction between protection and risk factors reduces the probability of a negative outcome [76]. Further, many of the studies included terms like hardiness, grit, mental toughness as equivalent terms for resilient performance, and also as factors that underpin resilience. The lack of theoretical foundation, and the complexity around resilience, has not prevented the appearance of tools designed to measure resilience in military samples.

**Measures and metrics used to study factors associated with resilient performance in defence and security settings.** *Measures of Performance*. There was a large degree of heterogeneity in measures used to assess performance in the included articles. Type of performance measure was categorised into four groups, these include:

1. Selection, Graduation, Completion, which assess participants based on a selection process (typically pass or fail).

2. Applied, including ecologically valid measures conducted 'in the field' such as land navigation or simulated captivity.

3. Shooting, usually conducted in a controlled environment collating performance scores. Although an ecologically valid measure of military performance, shooting performance was considered separately from those studies in the 'Applied' category.

4. Cognitive, including objective measures in a controlled setting such as sustained attention, memory or visual vigilance.

Studies identified in the review have been found to use one or more of these performance outcomes as outlined in Table 1.

*Measures of resilience*. Measures of resilience, which were included in the articles that met the inclusion criteria, were categorised into a) psychological measures, comprising subjective self-reports, and b) physiological measures, comprising objective biological measures. Measures utilised in all studies included in the review have been identified and briefly described alongside study characteristics in Table 1.

**Resilient performance in selection, graduation, completion (SGC) tasks.** Nine of the included studies assessed resilient performance in SGC tasks (see Table 1 for classification). Of these, five studies focused on selection and qualification courses. From these studies there is good evidence that mental toughness is important for successful completion of SGC tasks, and some evidence for the contribution of confidence, resilience (which authors define as a psychological construct) and a stress-is-enhancing mindset. Psychological hardiness may also be relevant, but the estimated effect size was small to medium on performance in these tasks, meaning that larger sample sizes are required to detect significant effects. Further, it may be the commitment facet of psychological hardiness that is specifically important for resilience in these performance paradigms that span a period of weeks. There is currently no consistent evidence for biological markers of resilience in SGC-type performance tasks. The use of similar constructs as predictors of resilient performance is noteworthy. Measures of resilience, grit, mental toughness and hardiness were found in the sample of literature reviewed. One difficulty

is that while resilience, grit, mental toughness and hardiness are proposed as different constructs, and each have unique measures, they may not be distinct. Indeed, a recent study testing content, construct and criterion validity of resilience, grit, hardiness, and mental toughness found substantial overlap among the four constructs [77]. Further, measures may not be sufficiently nuanced to provide a detailed overview of the processes in resilient performance. For example the Dispositional Resilience Scale-15 (DRS15; Bartone [78] measures hardy attitudes but does not include the hardiness process (i.e., hardy coping, hardy social support and hardy self-care; [79].

**Resilient performance in applied tasks.**   The stress-performance paradigms that were ecologically valid and conducted 'in the field' were grouped into the 'Applied' task category. Thirteen studies identified were categorised as Applied, of which nine utilised a captivity task (See Table 1).

Five studies measured biological markers of psychological resilience during captivity-type tasks. Markers included saliva cortisol and DHEA-S [68]. Overall performance during the high-intensity challenge was inversely related to the cortisol-DHEA-S ratio, whereas, performance during the low-intensity challenge was related to DHEA-S measured during the captivity phase. These findings are in contrast to other studies [63] that reported that the cortisol-DHEA-S ratio was significantly and positively associated with performance during the interrogation task. Lastly, measures of neuropeptide Y (NPY) [62] were associated with cortisol measures and interrogation performance. Those participants who had a larger cortisol and NPY response to interrogation stress, performed better during the interrogation. One final study did explore adrenal stress and physical performance during military survival training [67], however this did not include a measure of performance for the military task only a handgrip and vertical jump test.

Studies [33] also investigated the relationship between vagal tone (activity of the vagus nerve proposed as an index of emotion regulation and cognitive ability) and performance (captivity phase of the survival evasion resistance and escape (SERE) course and an Underwater Warfare Operations Combat Diver Qualification course). In all of these studies, reduced vagal tone was predictive of superior performance.

Three studies in the applied category assessed performance of military personnel in several physically and psychologically demanding tasks, which included, for example, timed runs, loaded marches, obstacle courses and land navigation (Table 1). The findings indicate that a stress-is-enhancing mindset was related to faster completion of the performance task, whereas a failure-is-enhancing mindset was related to a slower completion time than average [66]. Farina et al., [47] followed a large sample of 800 military personnel enrolled on the Special Forces Assessment and Selection (SFAS) course. Psychological (Intelligence quotient, Aptitude, Grit, Resilience) and biological (Cortisol, DHEA-S, testosterone, sex-hormone binding globulin (SHBG), and c-reactive protein (CRP)) markers of resilience were weakly correlated with the performance measures. Specifically, increases in DHEA-S were associated with poorer performance. Testosterone, adjusted for SHBG and adrenaline, were related to slower road march times while SHBG, adjusted for testosterone, was positively associated with performance on a number of outcomes. Lastly, increased noradrenaline was related with poorer performance. Biomarkers of stress and immune function were also investigated for their association with performance in 219 members of the Canadian Armed Forces [70]. Levels of cortisol, CRP, adiponectin, INF-γ, TNF-α, IL-1β, IL-2, IL-6, IL-8 and IL-18 were assessed alongside grip strength, aerobic capacity and performance on six military physical performance tests (sandbag fortification, escape to cover, picking and digging, picket and wire carry, stretcher carry and vehicle extrication). Higher CRP was associated with lower performance scores, while there was a positive association between IL-2 and grip strength, and adiponectin

was positively associated with plank time while negatively associated with grip strength. An important note here is that collectively these associations were all weak effects, and the multiple statistical analyses that were conducted increasing the chances of a false-positive result.

It is difficult to draw firm conclusions regarding markers of resilient performance in applied tasks given the range of settings, participants and methods used. Dissociative experiences consistently predict poorer performance in captivity tasks, and there is mixed evidence for the predictive value of cortisol, DHEA-S and NPY. Vagal reactivity also significantly predicted performance in captivity interrogation and in an underwater night-time navigation task several weeks after assessment, and therefore might be a useful marker of resilient performance. Other salivary and blood-based biomarkers produced varied predictive power of performance in these tasks. However, methodological limitations, such as small sample sizes and multiple statistical tests, could potentially explain these mixed findings.

**Resilient performance in shooting tasks.** Four studies assessed resilient performance in shooting tasks (See Table 1). One study explored the relationship between proinflammatory cytokine markers (TNF-a, IL-10), Brain-Derived Neurotrophic Factor (BDNF) and shooting performance in the elite combat unit of the Israel Defence Force [28]. Increases in inflammatory response were associated with lower levels of shooting performance; however, there was no relationship between the key marker of resilience, BDNF, and performance. The effect of sustained physical and psychological demand on marksmanship was explored during 'hell week' of Navy SEAL training [57]. Marksmanship was impaired during 'hell week' on a number of variables (e.g., distance from the centre of mass, shot group tightness, sighting time and number of missed targets). Performance on a range of cognitive variables (e.g., reaction time and accuracy on cognitive tasks) were worse, and mood was more negative, during 'hell week' compared to baseline. However, there was no analysis of how changes in key variables that might be indicative of greater resilience (e.g., more positive mood state) related to marksmanship performance.

Marksmanship performance in psychological demanding environments was the basis for a very well-controlled study of police officers [56]. Participants took part in shooting tasks in both high stress (24 trials) and low stress (24 trials) conditions (counterbalanced). Shooting accuracy was greater in the low stress compared with high-stress condition. Further analysis indicated officers with experience, high levels of thrill and adventure seeking, and low levels of behavioural inhibition performed better in the high stress setting, with greater experience being the strongest predictor.

The final study [45] explored how training programmes might be beneficial for shooting performance. Performance time on a shooting task was assessed in relation to participation in a training programme (Tactical Human Optimisation Rapid Rehabilitation and Reconditioning program; THOR3) that has aspects of psychological training. There was no statistically significant difference in performance between the participants who had undertaken the THOR3 programme and those who had not on performance time. Despite the difference not being statistically significant there was a small to medium effect for the difference between the groups, with better performance in the group who had undergone the THOR3 programme.

It is difficult to draw meaningful conclusions from four studies. However, the data does indicate that even among trained performers (e.g., Navy SEAL recruits and police offers) shooting performance under physical and psychological stress is impaired. From a resilience perspective, manipulating psychological stress and building on the work of Landman, Nieuwenhuys [56] shows promise. Introducing psychological stressors that draw on previous research such as uncertainty, or danger such as meaningful evaluation (psychological) or the potential for pain (physical), and effort (difficult task) will place participants under stress and enable exploration of resilience variables which may predict performance. It was also

noteworthy that aside from the Landman et al. study, there was no clear assessment of psychological and physiological state close to the task itself and while doing so presents challenges, this will give a clearer indication of what measures are important in the resilience-performance link.

**Resilient performance in cognitive tasks.** There were eight studies that collected data on cognitive performance (See Table 1). Four studies explored solely how cognitive performance deteriorated under psychological and physical stress [57–59,65], and four studies explored how markers of resilience may relate to cognitive performance [28,33,71,72].

Studies that explored cognitive performance deterioration used a range of computer based cognitive tasks following various psychologically demanding and stressful tasks including 'hell week' during NAVY seal training [57], a cold stress task [58], and SERE US Army training course [59]. All three studies indicated that the stress task or condition was associated with decreased cognitive performance. Różański, Jówko [65] explored the association between biochemical markers of stress and a divided attention cognitive task in male special force soldiers. There was an increase in performance on the divided attention task across a survival training programme. Further, while a trial test of the divided attention task was performed, there was no evidence that performance had plateaued and the increase in performance seen, despite the psychological and physical stress of the 48-hour exercise, may be representative of learning effects on this specific task. Collectively, while biomarkers of physical stress were taken in some, there was no exploration of how these variables changed in relation to performance on the cognitive task. As such, beyond concluding that the physical and psychological stress of a demanding environment has an impact on some aspects of cognitive performance, it is not possible to elucidate the resilience-performance link from the data.

Another four studies explored indicators of resilience and the link with cognitive function. Increases in heart rate [72] and BDNF [28] were associated with better cognitive function, suggesting promise for both as a marker of cognitive performance after a prolonged period of psychological and physical demand. However in other studies [71] serotonin levels were unrelated to cognitive performance and although vagal tone as explained previously predicted performance in an applied test and Navy SEAL selection, it was not associated with performance on a written test [33].

The data on cognitive performance reports relatively few studies that explicitly link markers of resilience with cognitive functioning. In the study by Yao, Yuan [72] it was interesting that a greater increase in heart rate was associated with better performance after errors during a task. That is, a stronger stress response (assessed by heart rate) was associated with better cognitive function, and is proposed to represent greater effort and mobilisation of cognitive resources to atone after the error. The finding that a stronger stress response may be associated with better recovery after an error is a timely reminder that a strong stress response is not, in and of itself, negative, and may have positive performance effects [80]. Similarly, given the sample of elite military performers, the data suggesting that following a period of psychological and physical demand (5-day training programme) BDNF concentration may be a promising biomarker of potential cognitive performance was also noteworthy [28].

**Programmes designed to bolster resilient performance in defence and security settings.** Surprisingly few programmes specifically described seeking to improve resilient performance in high-performance defence and security settings. Four studies explored cognitive performance in the context of training programmes [40,53,60]. There were, additionally, two studies that specifically utilised mental skills training to improve performance [29,32].

Three studies explored cognitive function in the context of mindfulness training programmes [54,60] and one using an attention training programme [40]. Jha and colleagues ran a series of studies aiming to promote cognitive resilience using mindfulness to protect against

the degradation of attention over time. In two separate studies performance remained more stable for those who received in class mindfulness training [53] and those who received training from Master Resilience Trainer [54] on a Sustained Attention to Response Task (SART). The study by Meland, Ishimatsu [60] explored mindfulness training with Norwegian military helicopter units during their preparation for redeployment to an aeromedical mission in a conflict area. Those who received mindfulness training reported lower perceived mental workload on the SART. However, there was no difference in performance accuracy and indeed the mindfulness training group reported significantly slower reaction times.

A range of other training programmes have also been investigated to increase resilient performance. This includes the study of training for Adaptability and Resilience in Decision-Making (STAR-DM) approach [40], in which groups of squad leaders who received the training were compared to control group squad leaders on their ability to make scenario based decisions. The small sample size (n = 13) means it is difficult to draw any firm conclusions, but there was tentative evidence that the STAR-DM training packages can improve decision-making during training. Another programme also included general mental skills training (MST; Jensen et al., [29], investigating the cognitive skills and performance across a 12-week, three stage, training course. Soldiers were randomly assigned to one of three specific groups embedded into the course: a training-as-usual group (n = 91, given free time during the structured MST instruction), general mental skills training, covering goal setting, arousal control, imagery, positive self-talk and focus/concentration (n = 47), and mindfulness-based mind fitness training (n = 65). Performance in both mental skills groups was better than the training-as-usual group during the phase one hike and phase three hike, although performance of the training-as-usual group was higher during the cognitively demanding communications test in phase 3, than either mental training group. Overall performance of the training-as-usual group was significantly the highest during phase 1, while performance of the general mental skills training group was significantly the highest during phase 2. Finally, while, cortisol levels were lowest in the mental training group pre-ambush, similar levels were seen across all three groups during the ambush and after the ambush itself. In sum, there is some evidence for mental training programmes enhancing cognitive performance and altering stress response prior to a stressful task, but the effects on actual performance were less clear.

A similar mental skills intervention was conducted by Fitzwater, Arthur [32] in male British Army Parachute Regiment recruits (P-Company) taking part in a training course. Five platoons were exposed to a psychological skills training program comprising goal setting, relaxation and arousal regulation, self-talk and imagery, while five platoons in the control condition did not receive any exposure to psychological skills training. There was a significant increase in the observer-rated measure of mental toughness in the intervention group and no change in the control group. In addition, individual performance was significantly higher during P-Company for the intervention group when controlling for fitness and leadership climate. In sum, the three-week intervention showed some positive effects, both in terms of changes in mental toughness and performance.

There is a paucity of research using interventions to improve performance resilience in defence and security settings. To date, focus has been on mindfulness or general psychological skills interventions. Both interventions show some promise, with effects generally clearer on cognitive tasks rather than in real world performance. The study by Fitzwater, Arthur [32] is noteworthy in that regard, with a clear measure of performance based on relevant tasks, and a clear focus on the potential mechanisms (e.g., psychological skills usage) while controlling for potential confounders (e.g., fitness level, leadership climate).

**Development of resilient performance framework.** To stimulate future work on resilient performance in defence and security settings, a theoretically informed organisation of factors,

integrating findings of the present review alongside principles identified in the broader resilient performance literature, is identified in Table 2.

Due to the diversity of research identified in the review, this is a logical and critical step for synthesising the existing literature around an agreed definition and understanding of what resilient performance is. As previously outlined in this review, resilient performance is defined as: The maintained or improved execution of competence under situational duress. The proposed framework includes three levels: global-contextual enablers, situational processes, and markers of resilient performance. Global-contextual enablers refer to the stable dispositional, trait and ability-like variables that were studied in a number of the reviewed articles (e.g., hardiness, mental toughness). These factors are referred to as 'resilient performance enablers', which is consistent with recent progress in the area of thriving under pressure [23]. In the broader literature researchers have used the term 'resilience factors' [81,82] to describe these same variables. However, this terminology is somewhat ambiguous given 'factors' is often used interchangeably with the term 'variables' to discuss any measured phenomenon. Stable resilient performance enablers are predicted to influence the situational stress response and onward resilient performance through situational 'resilient performance processes'. Such processes include physiological dynamics and psychosocial resources and appraisals that are activated in response to specific stressful demands.

Preliminary evidence identified in the review has implicated such processes in resilient performance, which is again consistent with the broader literature on stress, resilience and performance. As indicated earlier, although a number of key resilient performance processes were identified in the review, additional process variables (e.g., social connectedness) would also be expected to play an important role [27]. At the highest level are the specific markers of resilient performance. In the review, a number of indicators of performance relevant to defence and security were identified, including marksmanship, attentional focus, and land navigation,

**Table 2. Organisation of the resilient performance framework.** Developed based on the systematic literature review presented here, and principles identified in the broader resilient performance literature.

| Overarching factor | Variable categories | Example variables |
|---|---|---|
| **Resilient performance** | Physical | Behavioural persistence |
| | Technical | Marksmanship, skilled fine motor performance |
| | Cognitive | Vigilance, judgement, anxiety |
| | Team | Coordination, cooperation |
| **Resilient performance processes (situational)** | Physiological | Cortisol, DHEA, NPY, Oxytocin |
| | Psychological | Control, self-efficacy, challenge |
| | Social | Belonging, social connection |
| | Self-regulation | Reported dissociative symptoms |
| **Resilient performance enablers (global/contextual)** | Physical abilities | Fitness |
| | Cognitive abilities | Intelligence |
| | Experience | Service history |
| | Personality | Hardiness |
| | Motivation and values | Character strengths |
| | Coping | Flexibility |
| | Mental skills | Goal setting, imagery, mindfulness |
| | Leadership | Transformational Leadership |

Note: contents of the table were developed based on the systematic literature review presented here, and principles identified in the broader resilient performance literature.

amongst others. Broadly, situational indicators of resilient performance (i.e., those factors that indicate the likelihood of an individual being able to successfully complete a discrete task under conditions of adversity) can be considered along physical, technical, cognitive affective, and team lines. The specific variables that are indicative of resilient performance will be, in part, dictated by the tasks being completed. For instance, resilient performance during a direct-action mission will require performance across a range of skills that may be different to those required for completion of a reconnaissance operation [22]. Acknowledging the complexity of the human experience, resilient performance markers are expected to be networked in the way that Kalisch, Cramer [27] suggest, and so can exert interacting effects upon one another (e.g., physical performance can affect technical performance) and be closely coupled with the resilient performance processes. Experiences of resilient performance are likely to have a recursive relationship with psychological resources such as perceptions of control and self-efficacy. These recursive interactions where processes impact performance and vice versa, are expected to continue for the duration of stress exposure and may fluctuate depending on the changing nature of stressful demands.

## Discussion

The aim of the current review was to examine underpinning theories and models, measures and metrics, and approaches that could be used to maintain and train resilient performance in defence and security personnel. Overall, many of the studies lacked a clear theoretical underpinning. A variety of measures and metrics from self-report measures through to biomarkers were used to assess the associated antecedent factors that might be indicative of resilient performance. In that regard, these findings are similar to those of a recent meta-analysis on the link between psychological resilience and mental health and functioning in military personnel, where the heterogeneity of the measures of resilience, and health and functioning outcomes hindered clear conclusions [13]. The few training studies that were identified in the review focused on mental skills, mindfulness, and resilient decision-making, which might be targets for enhancing resilient performance. The following discussion is an attempt to integrate the findings of the review alongside more general progress related to the study of resilient performance as it pertains to defence and security settings.

In general, there was limited use of coherent theoretical perspectives to rationalise the purpose of the studies or the measurement practices undertaken and then discuss the study findings. Several of the articles did focus on well-established psychological principles (e.g., mental skills), however, these principles were typically not discussed or integrated into broader explanatory frameworks (resilience frameworks or otherwise). The lack of theory is problematic, especially when there are unexpected findings, like those that emerged in several of the reviewed papers (e.g., decreased vagal tone linked to optimal performance) [33]. To build on existing research and make progress in this area, there needs to be clarity around how resilience is defined, conceptualised and the expected relationships between important antecedents, processes and outcomes. A number of studies included in the present review treated resilience (explicitly or implicitly) as a trait-like variable. Although there continues to be debate about the nature of resilience [83], contemporary definitions and models of resilient performance specifically tend to view the construct more as a process of person-environment interactions, which can be influenced, but not directly represented, by underlying traits and dispositions [81]. The present work is guided by this latter view and the definition put forth by Adler, Williams [16] who suggested that resilience is the positive adaptation to adversity or stress. This positive adaption will support sufficient performance levels to maintain important competencies, like those identified by Pulakos, Arad [1], and achieve a successful outcome.

Future work in this area should draw on the resilience theory portrayed in contemporary general and performance psychology literature to provide sound studies of theory based on evidence-based expectations and measurement practices.

Unlike some other high-performance domains, exploring the resilience-performance association in defence and security settings is complex because performance can encompass such a wide range of tasks. For example, performance might include completing a physically demanding assault course and resisting interrogation in a prisoner of war scenario, through to conducting covert reconnaissance. The qualities needed to perform in these, and other, scenarios will be different., The dynamic resilience network approach [27], can be used to acknowledge these unique contextual features. As the model outlines psychological resilience as a variable construct, that is context specific and dependent on person-environment interactions. According to this perspective, resilience is best understood by the dynamics in the interconnections between physical, affective, cognitive and social nodes. Therefore, this is a useful starting point from which to develop an understanding of the resilience-performance relationship in defence and security settings. This framework is used to outline resilient performance enablers, resilient performance processes, and markers of resilient performance, and these are discussed below.

A number of resilient performance enablers, or trait-like resilience factors, were identified in the literature review. This included, amongst other variables, hardiness, grit and mental toughness. These enablers are expected to contribute to adaptive processes under conditions of stress and pressure, thus increasing the likelihood of resilient performances. There is evidence that these types of responses are likely to contribute to effective performances [84]. However, the current review identified unique enabling factors. There is an ongoing debate about the extent to which factors like hardiness, grit and mental toughness are divisible, or indeed whether they are all part of the same psychological construct [85]. Delineating enablers that have incremental predictive validity on resilient performance outcomes seems to be a worthwhile future pursuit. Looking beyond the current review, there are a range of other potential resilient performance enablers that might be of interest. Potential enablers include intelligence, personality, motivation and general perceptions of social support [20].

A variety of situational processes of resilience were identified in the literature. There was consistent evidence across specialist groups and general soldiers that during interrogation a higher level of self-reported dissociative symptoms was associated with poorer performance [46,62,69]. In contrast, there was no reliable, consistent evidence of biomarkers being associated with performance across a range of performance indicators. During captivity-type tasks reduced vagal tone was predictive of superior performance in the captivity phase of the SERE course [33]. In the same paper Morgan et al. reported that as with performance in the SERE course, reduced vagal tone was associated with improved performance in the underwater navigation task several weeks later. Other salivary and blood-based biomarkers (e.g., cortisol, adrenaline and DHEA-S) produced varied predictive power of performance in these tasks, although methodological limitations, such as small sample sizes and multiple statistical tests, could potentially explain these mixed findings. Outside of the military, recent studies examining cortisol and DHEA in performance settings indicate a relatively small contribution [24]. When the performance tasks were specifically focused on marksmanship or cognitive performance the results were less clear. One promising process biomarker is BDNF concentration [28] which was positively related to cognitive performance in an elite military sample.

From a process perspective, it was notable that there was limited consideration of social factors, specifically leadership and team factors (e.g., cohesion), in relation to performance. While leadership style was considered as a covariate by Fitzwater, Arthur [32], it did not feature in the remaining studies. There is consistent evidence that leaders, and leadership style, may

impact stress responses [86]. Further, social support has been consistently associated with positive stress responses and better performance under stress (see Meijen, Turner [26], for an overview). Satiation of relatedness needs (linked to leadership and social support) has also recently been associated with experiences of thriving under pressure [24]. Cohesive groups can help provide the support needed to fulfil social connection needs and in the process contribute to resilient function. A novel biomarker that may also help indicate the strength of social connection within a group and the potential buffering against stress is Oxytocin (a neuropeptide produced in the hypothalamus). Oxytocin plays an important role in prosocial behaviours [87] and is associated with lower levels of cortisol under acute stress [88–91], particularly in tasks that are sufficiently stressful to elicit a strong HPA axis response [92]. Further, oxytocin may be an important factor in determining a challenge state, which is considered to be an adaptive response to stress that is indicative of positive coping resources, in contrast to a threat state which reflects insufficient coping resources [26]. Meijen, Turner [26] also proposed that NPY might be associated with a challenge state, with peripheral plasma NPY (which was assessed in the military studies by Morgan and colleagues) having an effect of decreasing hypothalamic pituitary-adrenal (HPA) axis activation [93]. These overlapping processes are interesting and may point towards a collection of biomarkers that are directly implicated in the stress-resilience-performance relationship and should be explored in future work.

While there are some promising findings, there is no single process marker (biological, psychological or social) that has been consistently found to predict performance of defence and security personnel. This is not particularly surprising given that performance demands may vary across different defence and security populations. There are also methodological and practical challenges associated with collecting data from the number of personnel at the right times to be able to conduct robust and adequately powered studies. It could also be argued that the 'resilience' required to perform a physically demanding exercise is different to the 'resilience' required to work effectively as part of a team in a demanding environment. In the literature review, a range of performance contexts were identified: pass/fail, selection or a course, physical performance, applied performance, cognitive performance, marksmanship. These provide a useful starting point, but perhaps are not entirely inclusive of the range of competencies that may be relevant to the work of defence and security personnel, see Pulakos, Arad [1]. Future studies may conduct a more rigorous task analysis to capture the different physical, technical, cognitive and affective, and social dimensions of performance. It would be possible to examine in a systematic way, which processes (including biomarkers) are most predictive of performance in specific tasks.

One strand of research, from other performance domains, which may have utility in explaining a broad range of relevant performance outcomes is research around challenge and threat states. In addition to work highlighting the importance of situational challenge appraisals [20,21] there is a consistent body of work demonstrating different patterns of cardiac reactivity related to challenge and threat states [94,95]. A challenge state facilitates improved decision-making, effective and maintained cognitive function, decreased likelihood of reinvestment (which affects skills motor performance), efficient self-regulation, and increased anaerobic power; all of which are likely to lead to successful performance [25]. In contrast, a threat state leads to ineffective decision-making and cognitive function, increased likelihood of reinvestment (which impairs skilled performance), inefficient self-regulation, and decreased anaerobic power (compared to a challenge state); all of which are likely to lead to unsuccessful performance (Jones et al., 2009 [25]). In short, in a challenge state, Sympathetic-Adreno- Medullary (SAM) activation is fast-acting and represents the mobilisation of energy for action (fight or flight) and coping. A threat state accompanies slow-acting Pituitary-Adreno-Cortical (PAC) activity, (and SAM) activation and represents a "distress system" associated with

perceptions of actual harm [15]. As an underpinning philosophy, challenge and threat holds that stress itself is not deleterious for performance, rather, it is whether stress is characterised by challenge or threat that is important for performance. Building on this work related to cardiac responses under stress and merging it with some of the work on vagal tone may be a promising avenue.

The current review has wide implications for the development of resilient performance training. The resilient performance framework in Table 2 provides a context within which to understand how resilient-focused training programmes could be designed to develop and maintain or improve performance. Resilient-focused training programmes that were identified and reviewed were developed with a focus on mental skills, mindfulness and resilient decision-making. Given the small number of studies, caution is needed when interpreting and generalising the findings. However, there is initial evidence that training in some of these areas has a small-to-moderate effect on performance. Findings highlight why mental skills training, for example, might be the focus of emerging military research [96].

The link between mental skills and performance in the present work is consistent with the broader literature on the benefits of mental skills training in other performance settings, such as high-level sport [97]. The ability to use such skills appears beneficial for sustained performance under stress, yet the mechanisms of effect remain to be well-explored. This is especially true in defence and security contexts. Borrowing from theoretical frameworks related to stress self-regulation, there may be a dynamic component to the utilisation of these mental skills, which relies on an individual's capacity to accurately evaluate and appraise the demands of the context, select an effective strategy (i.e., a mental skill) and monitor the impact of that strategy on their performance, adjusting the approach as necessary [98]. It may be that this flexibility in the deployment of mental skills is equally, if not more, important than simply knowing about the skills themselves. This type of model provides the link between the knowledge and ability to use mental skills and the biopsychosocial processes that may be proximal determinants of resilient performance outcomes. Addressing these types of unknowns is important for developing the most effective training (mental skills or otherwise) for defence and security personnel.

Overall, based on the evidence that is available, training resilient performance in defence and security is likely to require a focus on multiple areas. The reviewed literature primarily emphasises targeting enablers (e.g., mental skills, mindfulness). However, earlier discussed correlational findings and results from beyond military populations also suggest that intervening at the process level, for instance by modifying resource appraisals and optimising stressor-strategy coupling, could reap performance benefits. In its delivery, training that includes educational, modelling and applied components should empower, coach and equip individuals and teams with the information and skills needed to foster resilient performance under conditions of stress and pressure. This training would potentially be most valuable when it is compatible and could be built into day-to-day activities such as marksmanship training, enabling ongoing opportunities for modelling and application, which may be beneficial for maintenance [96].

## Future directions

This is a growing area of research, which the current review has attempted to synthesise despite the lack of homogeneity in published articles. Of note is diversity in the sample, including papers that failed to meet the inclusion criteria of the current review (e.g., participants aged under 18, [99]. Additionally, a range of papers that did not formally measure performance outcomes yet the construct was part of the research context (for example Eid and Morgan [46].

Later work could seek to extend the scope of the current review to encompass a range of study types within VUCA settings. Several other suggestions for future work have been provided throughout the discussion, two of which are particularly important for the research topic. First, is the development of a valid and reliable measure (or collection of measures) of resilient performance in defence and security settings. Conducting rigorous, theoretically informed research, requires tools that can be used to accurately examine variables of interest. Given the organisation of factors in Table 2, any such tool designed to capture resilient performance in these settings is likely to need a dispositional and situational component and may incorporate various types of assessment method. In addition to measure development, systematically analysing the strength and direction of associations between enablers, processes and markers of performance over time is also a critical step. Approaching this work with clear hypotheses and conducting studies in an iterative fashion, would contribute to advancing understanding and lay the foundation for a biopsychosocial theory of resilient performance specific to defence and security personnel.

## Conclusions

Findings of the review suggest that although established psychological concepts were often discussed, studies were largely lacking in underpinning explanatory theory (resilience theories or otherwise). Furthermore, a range of performance-related measures were pinpointed, which may provide the basis for examining networks of resilient performance in future defence and security specific studies. In addition to performance markers, a variety of global/contextual resilient performance enablers and situational resilient performance processes were identified. These variables offer some indication as to the antecedents of resilient performance and potentially suggest a target for future resilient performance training. From a performance standpoint, there were some notable omissions in the literature. In particular, few studies examined the role of social factors in performance, despite this being a critical component in high performing teams. Existing training programmes were narrowly focused on mental skills, mindfulness and resilient decision making.

## Supporting information

**S1 Checklist.**
(DOCX)

**S2 Checklist.**
(DOCX)

## Acknowledgments

This work was funded by the Human Social Science Research Capability (HSSRC) research fund, project HS 1.025 Psychological Resilience to Maximise Human Performance.

## Author Contributions

**Conceptualization:** Marc Vincent Jones, Nathan Smith, Elizabeth Braithwaite.

**Data curation:** Marc Vincent Jones, Danielle Burns, Elizabeth Braithwaite.

**Formal analysis:** Nathan Smith, Danielle Burns, Elizabeth Braithwaite, Martin Turner, Andy McCann.

**Funding acquisition:** Marc Vincent Jones, Nathan Smith, Elizabeth Braithwaite, Martin Turner, Andy McCann.

**Methodology:** Marc Vincent Jones, Nathan Smith, Danielle Burns, Elizabeth Braithwaite, Martin Turner.

**Project administration:** Marc Vincent Jones, Nathan Smith.

**Supervision:** Marc Vincent Jones.

**Writing – original draft:** Marc Vincent Jones, Nathan Smith, Danielle Burns, Elizabeth Braithwaite, Martin Turner.

**Writing – review & editing:** Marc Vincent Jones, Nathan Smith, Martin Turner, Andy McCann, Lucy Walker, Paul Emmerson, Leonie Webster, Martin Jones.

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
