## [Decision Letter · Decision Letter 0]

8 Jul 2022

PONE-D-22-12214A Systematic Review of Resilient Performance in Defence and Security SettingsPLOS ONE

Dear Dr. Jones,

Thank you for submitting your manuscript to PLOS ONE. After careful consideration, we feel that it has merit but does not fully meet PLOS ONE’s publication criteria as it currently stands. Therefore, we invite you to submit a revised version of the manuscript that addresses the points raised during the review process.

We look forward to receiving your revised manuscript.

Kind regards,

Rogis Baker, Ph.D

Academic Editor

PLOS ONE

Journal Requirements:

2. Please note that in order to use the direct billing option the corresponding author must be affiliated with the chosen institute. Please either amend your manuscript to change the affiliation or corresponding author, or email us at plosone@plos.org with a request to remove this option.

Reviewers' comments:

Reviewer's Responses to Questions

**Comments to the Author**

1. Is the manuscript technically sound, and do the data support the conclusions?

Reviewer #1: Yes

Reviewer #2: Yes

Reviewer #3: Partly

2. Has the statistical analysis been performed appropriately and rigorously? 

Reviewer #1: N/A

Reviewer #2: I Don't Know

Reviewer #3: No

3. Have the authors made all data underlying the findings in their manuscript fully available?

Reviewer #1: Yes

Reviewer #2: No

Reviewer #3: Yes

4. Is the manuscript presented in an intelligible fashion and written in standard English?

Reviewer #1: Yes

Reviewer #2: Yes

Reviewer #3: Yes

5. Review Comments to the Author

Reviewer #1: The authors provide a systematic review of thirty-two papers relating attempts to measure psychological resilience of tactical personnel. This includes 1) theoretical frameworks for measuring resilience, 2) measures associated with resilience, 3) programs to improve resilience, and 4) the quality of the research related to the first three points.

With regards to each of the above points:

1) The authors were not able to make any clear conclusions due to the limited number of papers that included theory. As a whole, I don’t think this area adds much to the paper and may be worth removing.

2) This section I feel the authors were most successful with. While the outcomes are quite varied in terms of the measures being used, the authors did a good job clustering them into groups to attempt to make a cohesive statement.

3) Again, there were limited (n=6) and varied studies in this area, which makes it difficult to make strong conclusions, although they do see some concordance in the area of mindfulness and other psychological skill training.

4) The authors did a good job at rating the studies, as well as highlighting their strengths and weaknesses.

At the end while a applaud the authors attempt to develop a framework, I feel that one individual’s attempt may likely get lost in the shuffle, or even make things more disjointed. This may be best done as a consensus statement by a major military psychological society (e.g. APA Society for Military Psychology). This may be worth removing.

Other specific comments

- The process of citing a study by author is occasionally incorrectly done and confusing (e.g. Line 67). Please review and correct.

- There are a lot of leading phrases that are missing commas, which make the sentence difficult or confusing to read (e.g. Line 216, 231). Recommend review the paper carefully for clarity.

- Sentences on lines 208-211 don’t make sense

- Line 355: grit and resilience are capitalized but not the other factors

- Line 357: missing parenthesis

- Line 404-407: These statements seem to be contradictory to each other. How can there be no difference, but also a small to medium effect size? Please clarify

Reviewer #2: This paper highlighted systematic literature review to explore resilient performance in defence and security settings. The authors have tried to collect all possible information wrt resilience in defence performance. It is understood that there is no single biomarker that can be attributed to Physical, mental and cognitive performance. However, it seems some more highlights should be there regarding Biomarkers that can be connected to resilience of personnel in case of a security settings. This is very important because a biomarker can give a beforehand predictive information about a resilient performance that may be required in real scenario.

Also, a schematic representation of the search strategy used (including databases and literature review) will be helpful for better understanding of the methodology adopted in this study.

Reviewer #3: The review paper titled "A Systematic Review of Resilient Performance in Defence and Security Settings" has been exhaustively researched upon and covers an important issue in defence and security settings. However, my observations are as appended below:

1. The review paper is completely theoretical and can be made more comprehensive for the readers by using some statistical techniques ( charts , histogram etc.) by pooling the papers based on their subject and outcomes.

2. The review has not included research papers that have used various MCDM tools and techniques that have addressed the issues related to the resilient performance in defense and security.

3. The review has also not included research papers that have used various Human Reliability tools and techniques that have addressed the issues related to the resilient performance in defense and security.

4. The authors also need to explore more recent scientific work in this area.

5. The suggested work as an extension may be based on certain latest MCDM and HR model that are not available in the literature. This can strengthen the contribution of the authors.

6. In its present form, this paper can be definitely deemed a "research review" but it needs to be more scientific in terms of available scientific models, the examples of which have been provided at points 2 and 3 , that addresses such issues and can be further extended to model a number of factors that have been exhaustively deliberated in this paper.

6. PLOS authors have the option to publish the peer review history of their article (what does this mean?). If published, this will include your full peer review and any attached files.

Reviewer #1: No

Reviewer #2: No

Reviewer #3: **Yes: **Dr. Rajiv Nandan Rai

---

## [Author Response · Author response to Decision Letter 0]

28 Jul 2022

We are extremely grateful to the reviewers for their helpful comments. We have outlined how we have addressed their comments on the attached response to reviewers letter.

---

## [Editor Report · Decision Letter 1]

1 Aug 2022

A Systematic Review of Resilient Performance in Defence and Security Settings

PONE-D-22-12214R1

Dear Dr. Marc Jones,

We’re pleased to inform you that your manuscript has been judged scientifically suitable for publication and will be formally accepted for publication once it meets all outstanding technical requirements.

Kind regards,

Rogis Baker, Ph.D

Academic Editor

PLOS ONE
---

## [Editor Report · Acceptance letter]

9 Sep 2022

PONE-D-22-12214R1 

A systematic review of resilient performance in defence and security settings 

Dear Dr. Jones:

I'm pleased to inform you that your manuscript has been deemed suitable for publication in PLOS ONE. Congratulations! Your manuscript is now with our production department. 

Kind regards, 

on behalf of

Dr. Rogis Baker 

Academic Editor

PLOS ONE